# RNA-Binding Motif Protein 22 Induces Apoptosis via c-Myc Pathway in Colon Cancer Cells

**DOI:** 10.3390/molecules30061227

**Published:** 2025-03-09

**Authors:** Ye-Rin Park, So-Mi Park, Nanyeong Kim, Jihoon Jung, Seokwoo Kim, Kwan-Il Kim, Hyeung-Jin Jang

**Affiliations:** 1College of Korean Medicine, Kyung Hee University, 26, Kyungheedae-ro, Dongdaemun-gu, Seoul 02447, Republic of Korea; yerinp@khu.ac.kr (Y.-R.P.); thal5566@khu.ac.kr (S.-M.P.); knylike@naver.com (N.K.); johnsperfume@gmail.com (J.J.); kim66470@naver.com (S.K.); myhappy78@naver.com (K.-I.K.); 2Department of Science in Korean Medicine, Graduate School, Kyung Hee University, Seoul 02447, Republic of Korea; 3Division of Allergy, Immune and Respiratory System, Department of Internal Medicine, College of Korean Medicine, Kyung Hee University Medical Center, Kyung Hee University, 23, Kyungheedae-ro, Dongdaemun-gu, Seoul 02447, Republic of Korea

**Keywords:** RBM22, c-Myc, MID1IP1, CNOT2, colorectal cancer

## Abstract

RNA-binding motif 22 (RBM22) is an RNA-binding protein involved in gene regulation, with the capacity to bind DNA and function as a transcription factor for various target genes. Recent studies demonstrated that RBM22 depletion affects cell viability and proliferation of glioblastoma and breast cancer cells. However, the role of RBM22 in colon cancer and the molecular mechanisms underlying its tumor-suppressive function remain largely unclear. In this study, we demonstrate that RBM22 induces apoptosis and suppresses colon cancer cell viability and proliferation by modulating c-Myc expression. Furthermore, RBM22 knockdown reduces c-Myc stability. Therefore, our findings suggest that RBM22 depletion regulates cancer cell proliferation and induces apoptosis via the c-Myc pathway.

## 1. Introduction

Colorectal cancer (CRC) is the third deadliest and fourth most commonly diagnosed cancer worldwide, highlighting the urgent need for innovative therapeutic approaches [1]. While conventional cancer therapies such as chemotherapy, radiation, surgery, and immunotherapy have been utilized, recent studies emphasize the potential of molecular-targeted therapy in CRC treatment [2]. Hence, exploring novel genes and treatments to control cancer cells is imperative.

RNA-Binding Motif Protein 22 (RBM22) encodes an RNA-binding protein (RBP) primarily involved in pre-mRNA splicing. Recent studies have demonstrated the pivotal role of RBM22 in pre-mRNA splicing and revealed its involvement in various malignancies, reigniting interest in exploring its potential significance [3]. According to several studies, RBM22 was identified as a crucial factor for cancer cells and is overexpressed in triple-negative breast cancer and glioblastoma cells. The depletion of RBM22 reduces cell survival rates and inhibits both proliferation and metastasis [4,5,6]. However, the anti-cancer effects and underlying molecular mechanisms of RBM22 in colon cancer remain unclear.

The regulatory role of c-Myc in cancer cell proliferation, angiogenesis, and apoptosis through the modulation of diverse target genes is well established [7]. Numerous c-Myc target genes were identified, highlighting its critical involvement in signaling pathways that promote cancer cell growth [8]. c-Myc is essential for various fundamental cellular processes, such as ribosomal biogenesis, cell division, and survival, all of which are crucial for cancer cell growth [9]. Therefore, modulating c-Myc expression in cancer cells is crucial.

CCR4-NOT transcription complex subunit 2 (CNOT2) plays a pivotal role in apoptosis, autophagy, proliferation, and angiogenesis. Previous studies have indicated that the reduction in CNOT2 can enhance apoptosis, impede metastasis by suppressing proliferation, and inhibit angiogenesis [10,11]. MID1-interacting protein 1 (MID1IP1) is highly expressed across various cancer types, with its involvement in cancer cell growth and apoptosis evidenced by its association with c-Myc-mediated ribosomal proteins L5, L11, and CNOT2 [12]. Therefore, we investigated the association between c-Myc and RBM22, which is regulated by CNOT2 and MID1IP1, in CRC.

Our study revealed that RBM22 is involved in apoptosis and proliferation and is emerging as a novel regulator of other tumor-related genes. RBM22 modulates c-Myc through CNOT2 and MID1IP1, influencing c-Myc stability and contributing to a shortened half-life, thereby revealing its tumor-suppressive functions. Consequently, our findings suggest that RBM22 functions as a tumor suppressor by regulating cell survival and proliferation and inducing apoptosis via c-Myc.

## 2. Results

### 2.1. RBM22 Depletion Reduces Cytotoxicity and Proliferation in Colon Cancer Cells

We confirmed RBM22 expression and demonstrated its overexpression in colon cancer cells. RBM22 knockdown led to a time-dependent decrease in cell viability in HCT116^p53+/+^ and DLD-1 (Figure 1D). Additionally, RBM22 knockdown significantly reduced cell proliferation. These data indicated that RBM22 is overexpressed in colon cancer cells and is implicated in cell viability and proliferation. Given the greater decrease in cell viability and proliferation observed with RBM22 siRNA #2 compared to RBM22 siRNA #1, subsequent experiments were performed using RBM22 siRNA #2.

### 2.2. Knockdown of RBM22 Triggers Apoptosis

We assessed the ability of RBM22 depletion to induce apoptosis through annexin V/PI staining (Figure 1A). To determine the type of cell death, we treated the cells with a pan-caspase inhibitor of apoptosis (Z-VAD-FMK) or an autophagy inhibitor (3-methyladenine, 3-MA). Although the caspase inhibitor significantly rescued cell death-induced RBM22 depletion, the autophagy inhibitor exhibited no similar effect (Figure 2B), suggesting that RBM22 depletion might be associated with apoptosis rather than autophagy. To further confirm this, we investigated apoptosis markers and observed a significant decrease in PARP levels and an increase in cleaved-caspase3 (Figure 2C). These results indicate that RBM22 knockdown induces apoptosis in colon cancer cells.

### 2.3. RBM22 Is Regulated by Oncogene c-Myc

The overexpressed c-Myc in CRC, as is known from previous studies, was demonstrated to induce apoptosis through CNOT2 and MID1IP1 [13,14]. To investigate the relationship among RBM22 and c-Myc, CNOT2, and MID1IP1, we performed a correlation analysis using GEPIA. As shown in Figure 3A, RBM22 expression is positively correlated with c-Myc (R = 0.41, *p* < 0.01), CNOT2 (R = 0.52, *p* < 0.001), and MID1IP1 (R = 0.11, *p* = 0.0095), suggesting potential regulatory interactions. Therefore, we investigated whether CRC cells transfected with RBM22 siRNA regulated oncogenes, including c-Myc. When RBM22 was knocked down in CRC cells, no changes were observed in the RNA levels of MYC, CNOT2, or MID1IP1 (Figure 3B). However, decreased protein levels of c-Myc, CNOT2, and MID1IP1 were observed (Figure 3C).

### 2.4. RBM22 Modulates c-Myc Through CNOT2 and MID1IP1, Enhancing Apoptosis

The nuclear level of c-Myc decreased in cells transfected with RBM22 siRNA (Figure 4A). These data suggest that the deficiency of RBM22 in CRC cells regulates the protein levels of oncogenes and nuclear c-Myc. Previous studies have shown that the depletion of CNOT2 in HCT116 ^p53+/+^ cells enhances the antitumor effect of depleted MID1IP1, leading to a reduction in apoptosis-related factors, such as pro-caspase3, pro-PARP, and c-Myc [12]. Based on this, when CNOT2 or MID1IP1 was co-knocked down with RBM22, there was a more significant increase in cleaved-PARP and cleaved-caspase3, as well as a greater decrease in c-Myc, compared to samples in which RBM22 siRNA was transfected alone (Figure 4B,C). Furthermore, since ribosome biogenesis is essential for carcinogenesis, c-Myc is associated with the ribosomal protein L5. Therefore, we investigated whether the reduction in c-Myc expression induced by RBM22 knockdown is mediated by RPL5. The results showed that the decreased c-Myc expression observed when RBM22 was co-knocked down with RPL5 was reversed (Figure 4D). These results indicated that RBM22 knockdown induced apoptosis through CNOT2 and MID1IP1 and reduced c-Myc via RPL5.

### 2.5. RBM22 Attenuates c-Myc Stability and Sensitivity to Serum Stimulation

To investigate whether RBM22 knockdown regulates c-Myc stability, we used the DNA synthesis inhibitor CHX in HCT116 ^p53+/+^ cells. The results showed that, compared to the control siRNA, the half-life of c-Myc was significantly reduced when transfected with RBM22 siRNA (Figure 5A). These data indicated that when exposed to CHX, the group treated with RBM22 siRNA showed decreased c-Myc stability compared to the group treated with the control siRNA. As c-Myc rapidly responds to serum stimulation, 20% FBS was used to stimulate c-Myc expression in HCT116 ^p53+/+^ cells transfected with the control or RBM22 siRNA. The results showed that compared to the control siRNA-treated group, the expression of c-Myc was reduced in the RBM22 siRNA-treated group after 6 h (Figure 5B). These data suggested that RBM22 knockdown reduced the expression of serum-stimulated c-Myc.

## 3. Discussion

The formation and progression of colon cancer are associated with the dysregulation of RNA splicing factors [13]. RBM22, known for its role in pre-mRNA splicing, encodes RNA-binding proteins that can interact with multiple RNAs simultaneously and typically possess one or more RNA-binding domains [3].

The depletion of RBM22, that is overexpressed in CRC cells, resulted in a significant decrease in the viability and proliferation of HCT116 ^p53+/+^ and DLD-1 cells (Figure 1C,D). These findings suggest that RBM22 overexpression is associated with cancer cell growth and proliferation.

PARP plays a critical role in preventing apoptosis by detecting and repairing DNA damage [15]. Caspase3 is involved in the degradation of nuclear DNA during apoptosis, and its activation serves as a key executor of apoptosis, initiating cascade reactions [16]. Specifically, cleaved-caspase3 facilitates the cleavage of pro-PARP, resulting in its activation [17]. Apoptosis was confirmed in CRC cells transfected with RBM22 siRNA, as indicated by an increase in apoptotic cells through annexin V and PI staining (Figure 2A). Additionally, the apoptosis inhibitor Z-VAD-FMK, but not the autophagy inhibitor 3-MA, rescued cancer cell viability, demonstrating that RBM22 is involved in apoptosis rather than autophagy (Figure 2B). Furthermore, the regulation of apoptosis-related proteins, such as PARP and cleaved-caspase3, provides evidence that RBM22 regulates apoptosis (Figure 2C). Consequently, the regulation of RBM22 influences the occurrence of apoptosis in CRC cells.

c-Myc, that is overexpressed in various human cancers including CRC, has been extensively investigated as a potential target gene. Therefore, we investigated whether apoptosis resulting from RBM22 deficiency was associated with the c-Myc signaling pathway. The mRNA levels of CNOT2, MYC, and MID1IP1 did significantly differ from those in the control siRNA despite the effective reduction in RBM22 (Figure 3A). However, reduced RBM22 expression decreased the protein levels of c-Myc, CNOT2, and MID1IP1 (Figure 3B). This suggests that RBM22 regulates the expression of oncogenes at the translation level rather than the transcription level. Immunofluorescence staining confirmed that RBM22 deficiency led to a reduction in nuclear c-Myc expression (Figure 3C). Furthermore, RBM22 depletion reduced c-Myc stability by regulating its half-life during CHX treatment and demonstrated a rapid response to serum stimulation (Figure 5A,B). This study demonstrates that RBM22 mediates the apoptosis of CRC cells via the c-Myc signaling pathway.

Given that the depletion of CNOT2 or MID1IP1 induces apoptosis and these oncogenes are associated with c-Myc [18], we investigated whether c-Myc, regulated by RBM22, is associated with CNOT2 or MID1IP1. This study demonstrated that transfection with CNOT2 or MID1IP1 inhibited c-Myc expression in conjunction with RBM22 depletion. Based on previous findings that c-Myc regulates the expression of specific genes by binding to ribosomal proteins and that the deficiency of the ribosomal protein RPL5 upregulates c-Myc [9], we examined the association between RBM22 and RPL5 (Figure 4C). Our results indicate that the inhibition of c-Myc by RBM22 was rescued by RPL5 depletion, suggesting that RPL5 mediates the c-Myc suppression induced by RBM22 deficiency.

In conclusion, we demonstrated that RBM22 regulates oncogenes, cell viability, and proliferation through c-Myc in CRC cells, suggesting that RBM22 may function as a novel oncogene. Therefore, RBM22 is expected to be used for treatment as a direct target in colon cancer, and research on related inhibitors is needed.

However, further research, including in vivo studies, is required to elucidate the anti-cancer mechanisms of RBM22 in CRC. Future research should specifically examine whether RBM22 directly interacts with CNOT2 and MID1IP1, thereby influencing their expression and apoptosis.

## 4. Materials and Methods

### 4.1. Chemicals and Reagents

Z-VAD-FMK (HY-15763) and 3-methyladenine (HY-19312) were purchased from MedChemExpress (Monmouth Junction, NJ, USA).

### 4.2. Cell Culture and Gene Silencing Using Small Interfering RNA (siRNA)

HCT116 ^p53+/+^ and DLD-1 cells were purchased from the Korean Cell Line Bank (KCLB; Seoul, Republic of Korea). The cells were cultured in Roswell Park Memorial Institute (RPMI)-1640 medium supplemented with 10% fetal bovine serum and 1% antibiotics. All cells were cultured at 37 °C in a 5% CO_2_ humidified atmosphere. The cells were seeded on a plate overnight and transfected with the control, RBM22, CNOT2, and MID1IP1 siRNAs (Bioneer, Daejeon, Republic of Korea) using INTERFERin (Polyplus-transfection SA, Illkirch-Graffenstaden, France), as described previously [19]. The cells were harvested after 72 h for subsequent experiments.

### 4.3. Cell Viability Assay

The cells were seeded in 96-well plates (5 × 10^3^ cells/well) and transfected with siRNA. Cell viability was determined using 3-(4,5-dimethylthiazol-2-yl)-,5-diphenyltetrazolium bromide (cat no. M6494) (MTT; Sigma-Aldrich Co., St. Louis, MO, USA). After incubation, the formazan was dissolved in dimethyl sulfoxide (cat no. D0458) (DMSO), and the absorbance was measured at 540 nm using a microplate reader (Bio-Rad, Hercules, CA, USA).

### 4.4. Colony Formation Assay

The cells were transfected with RBM22 for 72 h and trypsinized. The cells were seeded in six-well plates (1 × 10^3^ cells/well) and cultured for 1 week at 37 °C in a 5% CO_2_. Following colony formation, the cells were fixed with methanol and stained using a Diff-Quick Kit (Sysmex Corporation, Kobe, Hyogo, Japan) for 30 min.

### 4.5. Annexin V/Propidium Iodide (PI) Assay Using Flow Cytometry

The transfected cells were trypsinized and washed with phosphate-buffered saline (PBS). Subsequently, the cells were resuspended in a binding buffer containing fluorescein isothiocyanate (FITC)-tagged annexin V and PI. The apoptosis rate was assessed using a FACSCanto II flow cytometer (BD Biosciences, Becton-Dickinson, Franklin Lakes, NJ, USA).

### 4.6. Western Blotting

The cells were harvested and lysed in lysis buffer (Cell Signaling Technology, Beverly, MA, USA). Equal amounts of protein samples were separated by sodium dodecyl sulfate-polyacrylamide gel electrophoresis (SDS-PAGE) and subsequently transferred onto nitrocellulose membranes. After transfer, the membranes were incubated with primary antibodies, including Poly(ADP-ribose) polymerase (PARP)(cat no. 9542S), cleaved-PARP(cat no. 5625S), cleaved-caspase3(9661S), CNOT2(cat no. 34214S) (Cell Signaling Technology, Beverly, MA, USA), RBM22(cat no. 22103-1-AP), MID1IP1(cat no. 15764-1-AP) (ProteinTech Antibody Group, Chicago, IL, USA), c-Myc(cat no. ab32072) (Abcam, Cambridge, UK), and β-actin(cat no. sc-47778) (Santa Cruz Biotechnology, Dallas, TX, USA). Antibodies were diluted 1:1000 in tris-buffered saline with 0.1% Tween 20 and incubated at 4 °C. Horseradish peroxidase-conjugated secondary antibodies (cat no. sc-516102 and cat no. sc-2004) were then added and incubated for 1 h. The protein bands were visualized using ImageQuant LAS 500 (GE Healthcare Life Sciences, Sydney, Australia).

### 4.7. Immunofluorescence

The cells were seeded onto two-well culture slides at a density of 1 × 10^5^ cells per well and transfected with RBM22 for 72 h. Subsequently, the cells were fixed with 4% paraformaldehyde and permeabilized with 0.2% Triton X-100 for 15 min. The cells were then incubated overnight at 4 °C with an antibody against c-Myc (1:200; Abcam, Cambridge, UK), followed by incubation with Alexa Fluor 488 goat anti-rabbit IgG antibody (1:500; Invitrogen, Waltham, MA, USA) and incubated for 1 h. The nuclei were stained using 4,6-diamidino-2-phenylindole (DAPI), and images were captured using the CELENATM S Digital Imaging System (Logos Biosystems, Inc., Anyang-si, Gyeonggi-do, Republic of Korea).

### 4.8. Real-Time Polymerase Chain Reaction (RT-PCR)

RNA extraction was performed using Hybrid-RTM (product no. 301-101, GeneAll, Seoul, Republic of Korea). RNA concentration was measured using a NanoDrop spectrophotometer (Thermo Fisher Scientific, Waltham, MA, USA). Total RNA was reverse-transcribed into c-DNA using Maxime RT premix (product no. 25082, iNtRON Biotechnology, Seongnam-si, Republic of Korea). Real-time PCR was conducted using an Applied Biosystems Step One System (Applied Biosystems, Foster City, CA, USA) with Universal SYBR Green Master Mix (product no. 4367659, Applied Biosystems, Foster City, CA, USA). To quantify mRNA expression, the relative expression of the target gene was calculated relative to GAPDH (2^−∆∆Ct^). Primers for *MYC*, *CNOT2*, and *MID1IP1* were purchased from Bioneer (Daejeon, Republic of Korea).

### 4.9. Cycloheximide (CHX) Chase Assay

The cells were seeded and transfected into six-well plates (7 × 10^4^ cells/well). The protein synthesis inhibitor CHX (50 µg/mL) was subsequently added at 0, 30, 60, and 90 min. Protein expression was performed to verify protein expression.

### 4.10. Serum Stimulation for c-Myc Induction

Transfected cells were seeded in six-well plates (7 × 10^4^ cells/well) and starved in serum-free medium for 24 h. The cells were then incubated with 20% FBS and harvested at 0, 6, 12, and 24 h. Protein expression levels were then assessed using Western blotting.

### 4.11. Statistical Analysis

The data were replicated at least thrice and are presented as the mean standard ± deviation (SD). One-way analysis of variance (ANOVA) followed by Dunnett’s test was performed for multiple groups, and Student’s *t*-test was used to compare two groups. Statistical significance was determined using the GraphPad Prism software (version 8.0; San Diego, CA, USA).

## 5. Conclusions

In summary, our findings reveal that RBM22 functions as a tumor suppressor in colon cancer by regulating the c-Myc stability, a key factor in cancer cell proliferation and survival. Silencing RBM22 results in a significant reduction in c-Myc expression, accompanied by increased apoptosis and decreased viability of colon cancer cells. Our study demonstrates the potential of RBM22 as a therapeutic target for modulating c-Myc-driven oncogenesis in colon cancer.

## Figures and Tables

**Figure 1 molecules-30-01227-f001:**
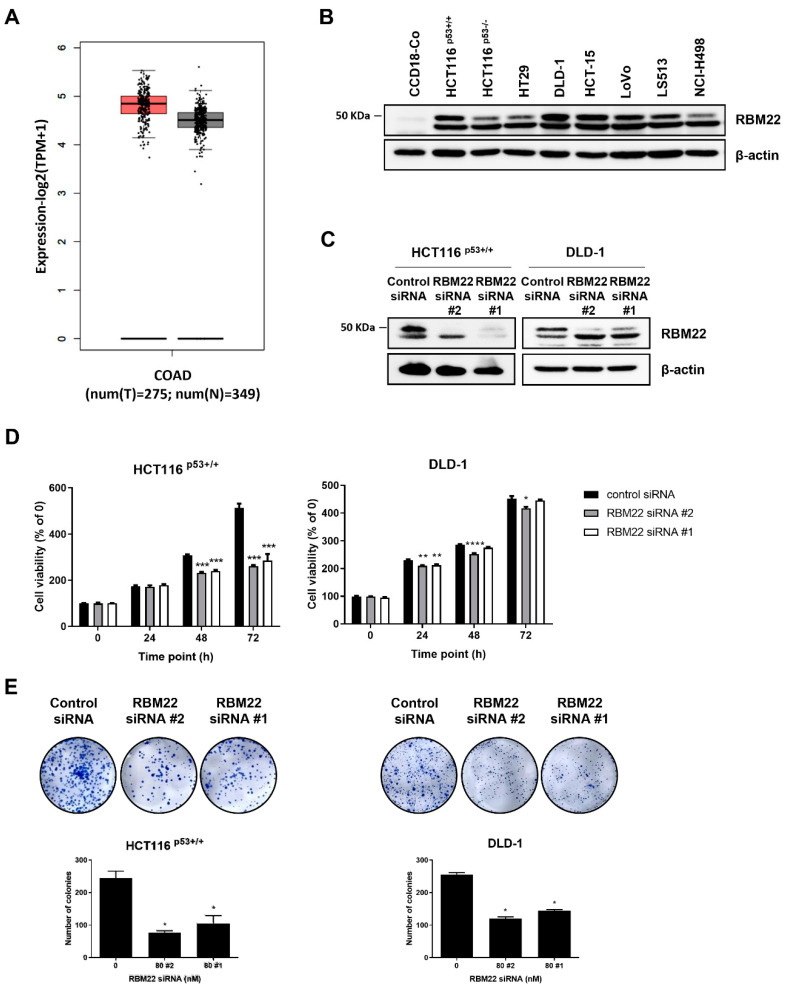
RBM22 enhances cancer cell viability and proliferation. (**A**) RBM22 expression in tumor (red) and normal (black) in COAD. (**B**) Expression levels in CRC cells of RBM22. (**C**) Expression of RBM22 in HCT116 ^p53+/+^ and DLD-1 cell lines with siRNA #1 and #2 transfection. (**D**) Time-dependent effects of RBM22 depletion on cell survival rates. (**E**) Colony formation in CRC cells and quantification. Data (n ≥ 3) are represented as mean ± SD. * *p* < 0.05, ** *p* < 0.01, *** *p* < 0.001, and **** *p* < 0.0001.

**Figure 2 molecules-30-01227-f002:**
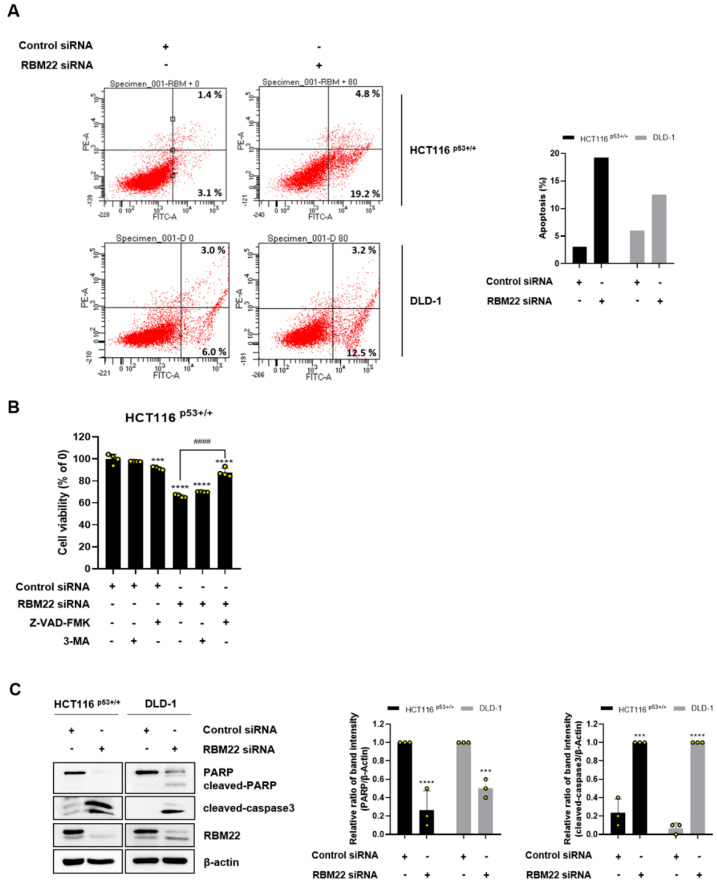
RBM22 regulates apoptosis in CRC cells. (**A**) To evaluate cell death rate associated with RBM22 depletion, annexin V/PI staining was conducted, followed by analysis using flow cytometry. (**B**) MTT assay was performed by transfecting cells with RBM22, an apoptosis inhibitor (20 μM, Z-VAD-FMK), and an autophagy inhibitor (5 mM, 3-MA). (**C**) Protein expressions of PARP and cleaved-caspase3 following RBM22 depletion and quantification. Data (n ≥ 3) are represented as mean ± SD. *** *p* < 0.001, **** *p* < 0.0001 compared to control group. ^####^ *p* < 0.0001 indicates significant results compared to RBM22 knockdown group only.

**Figure 3 molecules-30-01227-f003:**
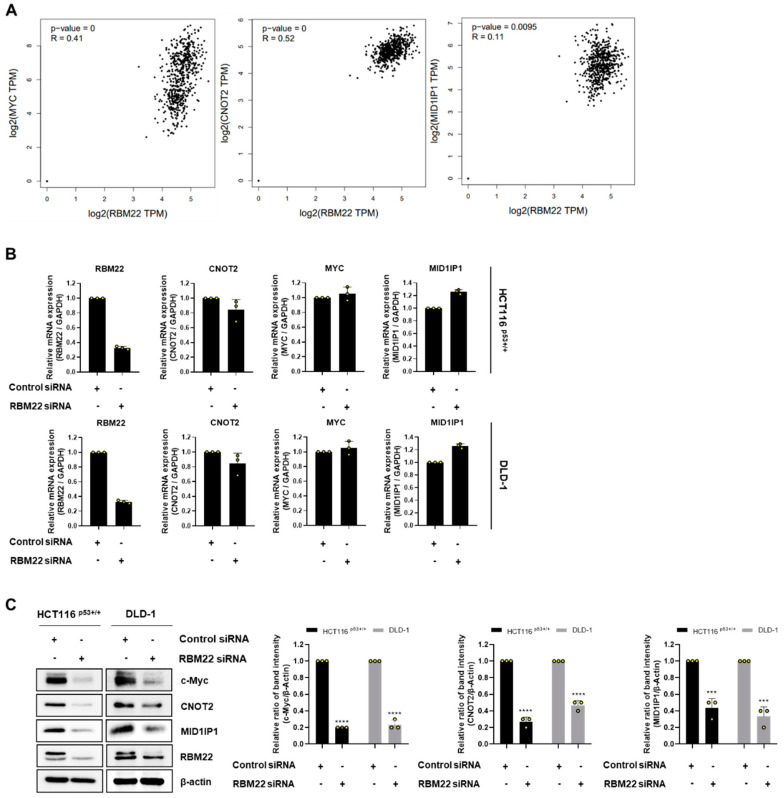
RBM22 regulates c-Myc through CNOT2 and MID1IP1. (**A**) Correlation analysis among RBM22 expression and c-Myc, CNOT2, and MID1IP1 expression. Pearson’s correlation was used to assess relationships, and correlation coefficients and *p*-values are indicated on each plot. Expression levels are presented as log2-transformed transcripts per million (TPM) values. (**B**) mRNA expression of RBM22, CNOT2, MYC, and MID1IP1 followed by RBM22 depletion. (**C**) Protein expression of c-Myc, CNOT2, and MID1IP1 following RBM22 depletion. Data (n ≥ 3) are represented as mean ± SD. *** *p* < 0.001, **** *p* < 0.0001 compared to control group.

**Figure 4 molecules-30-01227-f004:**
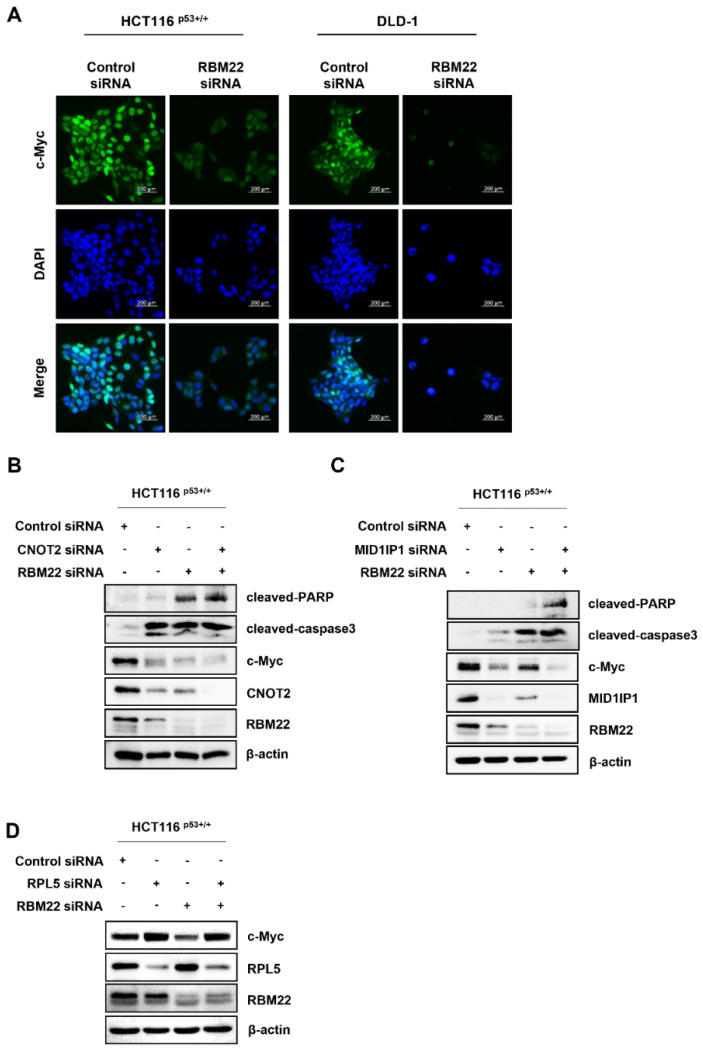
Impact of ribosomal protein L5 on c-Myc expression with RBM22. (**A**) Fluorescence expression of c-Myc in CRC cells. Enhanced apoptosis via CNOT2 or MID1IP1 depletion. (**B**) Effects on apoptosis and oncogene related protein expressions following combination of RBM22 and CNOT2 siRNA. (**B**,**C**) Effects on apoptosis and oncogene related protein expressions following combination of RBM22 and MID1IP1 siRNA. (**D**) Effect of RBM22 on RPL5 protein expression.

**Figure 5 molecules-30-01227-f005:**
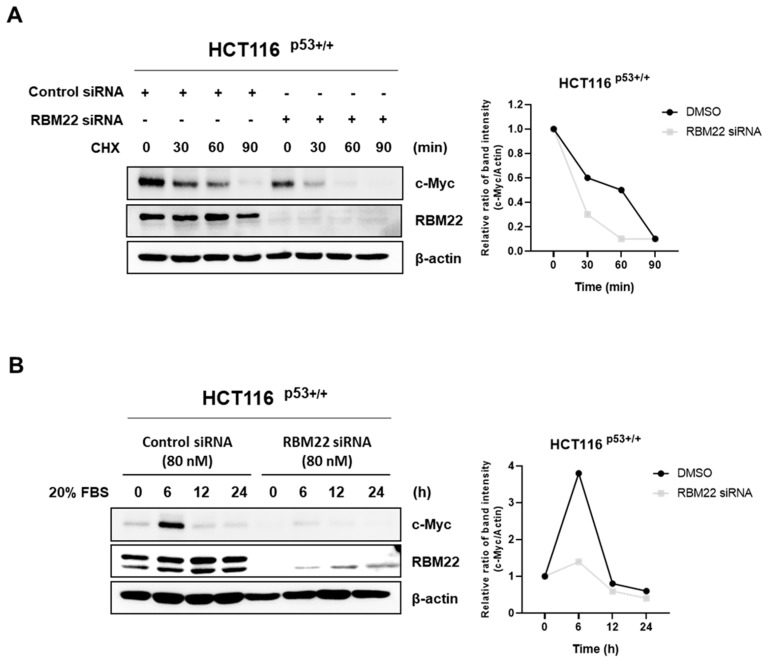
RBM22 regulates c-Myc stability and serum stimulation. (**A**) c-Myc stability assessed using CHX. c-Myc protein level declined more rapidly in RBM22 siRNA-treated cells than in control siRNA-treated cells, suggesting that RBM22 knockdown accelerates c-Myc degradation. (**B**) Regulation of serum induced activation of c-Myc.

## Data Availability

Data are contained within the article.

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
