# Peer review of "RNA-Binding Motif Protein 22 Induces Apoptosis via c-Myc Pathway in Colon Cancer Cells"

_molecules, 2025, doi:10.3390/molecules30061227_

Round 1
Reviewer 1 Report
Comments and Suggestions for Authors
This is a very interesting and well-executed empirical study on exploring the role of RNA-Binding Motif Protein 22 (RBM22) in colon cancer, focusing on its involvement in inducing apoptosis and regulating cancer cell proliferation through the c-Myc pathway. There are some question to clarify the results and ensure the findings are robust, reproducible, and relevant for future research in the field.
Specific Comments
1) Would you provide more detailed mechanistic insights into how RBM22 interacts with these factors at the molecular level? Specifically, does RBM22 physically interact with c-Myc or its regulators, or is the effect more indirect?
2) In the apoptosis experiments (Annexin V/PI staining), the authors mention the use of Z-VAD-FMK as an apoptosis inhibitor. Was this used as a positive control in the experiment? It would be helpful to include a discussion of other possible mechanisms through which RBM22 might influence apoptosis, apart from its interaction with CNOT2 and MID1IP1.
3) In the apoptosis experiments (Annexin V/PI staining), you mention the use of Z-VAD-FMK as an apoptosis inhibitor. Was this used as a positive control in the experiment? It would be helpful to include a discussion of other possible mechanisms through which RBM22 might influence apoptosis, apart from its interaction with CNOT2 and MID1IP1.
4) This manuscript suggests that RBM22 could be a novel therapeutic target. In terms of drug development, what specific inhibitors or activators of RBM22 do have you foresee being potentially useful in a therapeutic context for CRC? Additionally, would you recommend targeting RBM22 directly, or would indirect modulation through its downstream pathways be more feasible?
Author Response
Feb 27, 2025
Dear Chief Editor
Resubmission of Manuscript ID: molecules-3455452
We would like to thank you for the letter dated 20th Feb, 2025, and we hereby resubmit a revised copy of the manuscript for your consideration. We would also like to express our thanks to you and all the reviewers for insightful comments that have greatly helped improve the quality of the manuscript.
In this manuscript, we have studied carefully and revised and added details, reflecting the overall advice of previous reviewers. We hope that we addressed well to comments and also wish our revised manuscript would be accepted for publication in Molecules.
Best regards, Hyeung-Jin JangReviewer #1:
Dear Reviewer,
We sincerely appreciate your time and effort in reviewing our manuscript. Your valuable insights and constructive feedback have significantly contributed to improving the quality of our work. We have carefully addressed all the comments and revised the manuscript accordingly.
Thank you for your thoughtful review and consideration.
Best regards,
Hyeung-Jin Jang
Specific Comments
Point 1) Would you provide more detailed mechanistic insights into how RBM22 interacts with these factors at the molecular level? Specifically, does RBM22 physically interact with c-Myc or its regulators, or is the effect more indirect?
(Response) We sincerely appreciate the reviewer’s insightful question. The direct physical interaction between RBM22 and c-Myc or its associated regulators has not been experimentally confirmed. Our results suggest that RBM22 may regulate c-Myc stability rather than its transcription, possibly through post- translational modifications or interaction with key regulatory proteins such as CNOT2 and MID1IP1. To further clarify this mechanism, we have now included a statement in the Discussion section (lane 208-210) acknowledging the need for additional studies to determine whether RBM22 directly interacts with c-Myc and its regulators or exerts its effects through other molecular pathways.
Point 2, 3) In the apoptosis experiments (Annexin V/PI staining), the authors mention the use of Z-VAD-FMK as an apoptosis inhibitor. Was this used as a positive control in the experiment? It would be helpful to include a discussion of other possible mechanisms through which RBM22 might influence apoptosis, apart from its interaction with CNOT2 and MID1IP1.
(Response) We sincerely appreciate the reviewer’s insightful comments and the opportunity to clarify our experimental approach. In the Annexin V/PI staining experiments, Z-VAD-FMK was not used; rather, only RBM22 siRNA knockdown was performed to assess apoptosis induction. Furthermore, Z-VAD-FMK was not employed as a positive control but rather as an apoptosis inhibitor to investigate whether inhibiting apoptosis could rescue the decrease in cell viability observed upon RBM22 knockdown. We recognize the importance of further investigating these potential mechanisms, and we would now like to conduct further studies to explore these pathways. We hope this clarification adequately addresses the reviewer’s concerns, and we appreciate the valuable feedback that has helped us refine our manuscript.
Point 4) This manuscript suggests that RBM22 could be a novel therapeutic target. In terms of drug development, what specific inhibitors or activators of RBM22 do have you foresee being potentially useful in a therapeutic context for CRC? Additionally, would you recommend targeting RBM22 directly, or would indirect modulation through its downstream pathways be more feasible?(Response) We sincerely appreciate the reviewer’s insightful question regarding the therapeutic potential of RBM22 and the feasibility of direct versus indirect targeting strategies. The compound anthraquinone-2- sulfonic acid (AQ2S) has been reported to act as a competitive inhibitor of the RanBP2-type zinc finger within the RNA Binding Motif (RBM) protein domain, thereby disrupting the binding of small RNAs. (Jackson and Kochanek, 2020) Additionally, AQ2S has been proposed as a Caspase-3 inhibitor and has also been suggested as a RBM inhibitor. (Kochanek, 2017) So, further studies are planned to identify additional RBM22 inhibitors and to investigate the potential effects of RBM22 inhibitor on CRC. Understanding how RBM22 inhibitor interacts with RBM22 and its downstream pathways may provide valuable insights into the feasibility of RBM22 as a therapeutic target. Furthermore, if the RBM22 inhibitor and mechanisms are revealed, we anticipate that direct targeting of RBM22 rather than downstream pathways would be more effective in the treatment of CRC. We greatly appreciate the reviewer’s thoughtful input, which has helped us refine our manuscript.
Reference
JACKSON, T. C. & KOCHANEK, P. M. 2020. RNA Binding Motif 5 (RBM5) in the CNS-Moving Beyond Cancer to Harness RNA Splicing to Mitigate the Consequences of Brain Injury. Front Mol Neurosci, 13, 126.
KOCHANEK, T. C. J. C. V. 2017. Small molecule inhibitors of rna binding motif (rbm) proteins for the treatment of acute cellular injury.

Reviewer 2 Report
Comments and Suggestions for Authors
Title : RNA-Binding Motif Protein 22 Induces Apoptosis via c-Myc 2 Pathway in Colon Cancer Cells
In this paper, Ye-Rin Park and associates. describes how RNA-binding motif protein 22 contributes to the development of colorectal cancer and demonstrates how low expression levels of RBM22 cause apoptosis and inhibit the viability of cancer cells by modifying the stability of c-MYC. They discovered that cycloheximide treatment decreased c-MYC stability, especially in the RBM2 knockdown state. However, this MS. does not explain the molecular mechanism that explains decreased c-myc stability in reduced levels of RBM22. On the other hand, c-MYC stability is decreased in Figure 5A without affecting RBM2 values, indicating that RBM2 is not directly involved in c-MYC protein stability. The authors ought to carry out more experiments to support their assertions and revise the text as necessary.
Major comments:
Authors should elaborate results for each figure.
Authors should provide more details for experimental methods, cell viability assay.
For many chemicals and antibodies, catalogue numbers are missing
Minor comments on MS.
References are missing in the introduction, particularly at lane 30.
Rewrite the sentence, grammar mistake in lane 257.
In lane, Western blotting analysis performed to verify protein expression?? Change the expression to stability.
Author Response
Feb 27, 2025
Dear Chief Editor
Resubmission of Manuscript ID: molecules-3455452
We would like to thank you for the letter dated 20th Feb, 2025, and we hereby resubmit a revised copy of the manuscript for your consideration. We would also like to express our thanks to you and all the reviewers for insightful comments that have greatly helped improve the quality of the manuscript.
In this manuscript, we have studied carefully and revised and added details, reflecting the overall advice of previous reviewers. We hope that we addressed well to comments and also wish our revised manuscript would be accepted for publication in Molecules.
Best regards, Hyeung-Jin JangReviewer #2:
Dear Reviewer,
We sincerely appreciate your time and effort in reviewing our manuscript. Your valuable insights and constructive feedback have significantly contributed to improving the quality of our work. We have carefully addressed all the comments and revised the manuscript accordingly.
Thank you for your thoughtful review and consideration.
Best regards,
Hyeung-Jin Jang
Major comments:
Point 1) Authors should elaborate results for each figure.
(Response) As the reviewer pointed out, we have enhanced the manuscript's coherence and rigor by providing more detailed explanations of the results. The revised manuscript now offers a clearer interpretation of each figure and its significance. We appreciate the reviewer’s meticulous review.
Point 2) Authors should provide more details for experimental methods, cell viability assay.
(Response) As advised by the reviewer, additional details regarding the cell viability assay have been included in the Methods section (lane 229-230) to improve clarity and reproducibility.
Point 3) For many chemicals and antibodies, catalogue numbers are missing
(Response) Thank you for your opinion so that the quality of the paper can be improved. We have carefully reviewed the manuscript and added the missing catalogue numbers for chemicals and antibodies where applicable in the Methods section (lane 248-254).
Minor comments on MS.
Point 4) References are missing in the introduction, particularly at lane 30.
(Response) As the reviewer recommended, we have added the appropriate references to support the statements in the introduction.
Point 5) Rewrite the sentence, grammar mistake in lane 257.
(Response) As the reviewer recommended, we have revised the sentence to improve grammatical accuracy (lane 268).
Point 6) In lane, Western blotting analysis performed to verify protein expression?? Change the expression to stability.
(Response) Thank you for your valuable suggestion. We have modified the expression accordingly to accurately reflect protein stability, improving the clarity of the manuscript (lane 281).

Round 2
Reviewer 2 Report
Comments and Suggestions for Authors
The revised manuscript responded to most of the comments and improved the MS for publication. However, the authors have not explained the molecular mechanism that explains the reduced c-myc stability with reduced RBM22 levels, which is the main claim of the article. Figure 5A The stability of c-MYC is reduced without significant changes in RBM2 levels suggesting that RBM2 is not directly involved in the stability of c-MYC protein. The authors did not include most of the agent and antibody catalogues.
Author Response
Dear Reviewer,
We sincerely appreciate your time and effort in reviewing our manuscript. Your valuable insights and constructive feedback have significantly contributed to improving the quality of our work. We have carefully addressed all the comments and revised the manuscript accordingly.
Thank you for your thoughtful review and consideration.
Best regards,
Hyeung-Jin Jang
Comments and Suggestions for Authors
The revised manuscript responded to most of the comments and improved the MS for publication. However, the authors have not explained the molecular mechanism that explains the reduced c-myc stability with reduced RBM22 levels, which is the main claim of the article. Figure 5A The stability of c-MYC is reduced without significant changes in RBM2 levels suggesting that RBM2 is not directly involved in the stability of c-MYC protein. The authors did not include most of the agent and antibody catalogues.
(Response) We appreciate the reviewer’s insightful comments regarding the molecular mechanism underlying c-Myc stability in response to RBM22 knockdown. CHX is a well-established protein synthesis inhibitor used to assess the degradation rate of pre-existing proteins. In Figure 5A, we performed a CHX chase assay in HCT116 p53+/+ cells to compare c-Myc protein stability between control siRNA and RBM22 siRNA-transfected conditions.
Western blot analysis shows that RBM22 depletion resulted in decreased c-Myc stability compared to control siRNA. The quantification graph indicates that the relative intensity of c-Myc normalized to β-actin decreased faster in the RBM22 siRNA-transfected conditions than in the control. However, in control siRNA, RBM22 protein levels remained largely unchanged upon CHX treatment, suggesting that RBM22 is not rapidly degraded. This indicates that RBM22 is more likely involved in regulating c-Myc stability rather than being rapidly degraded. The CHX treatment time will be extended in further studies to determine whether RBM22 undergoes degradation at later time points. These findings suggest that RBM22 knockdown accelerates c-Myc degradation, supporting our hypothesis that RBM22 contributes to c-Myc stability.
We have revised the figure legend to explicitly state that RBM22 knockdown reduces c-Myc half-life, supporting our main claim that RBM22 is involved in the regulation of c-Myc stability. However, the precise molecular mechanism by which RBM22 regulates c-Myc stability remains unclear. Therefore, further studies will be conducted to investigate whether RBM22 affects c-Myc translation, protein-protein interactions, or specific degradation pathways.
Additionally, we have updated the Materials and Methods section to include the catalog numbers for all antibodies and chemical reagents used in the study.
